# FINE-GRAINED LOCAL SENSITIVITY ANALYSIS OF STANDARD DOT-PRODUCT SELF-ATTENTION

## ABSTRACT

Self-attention has been widely used in various machine learning models, such as vision transformers. The standard dot-product self-attention is arguably the most popular structure, and there is a growing interest in understanding the mathematical properties of such attention mechanisms. This paper presents a fine-grained local sensitivity analysis of the standard dot-product self-attention. Despite the well-known fact that dot-product self-attention is not (globally) Lipschitz, we develop new theoretical local bounds quantifying the effect of input feature perturbations on the attention output. Utilizing mathematical techniques from optimization and matrix theory, our analysis reveals that the local sensitivity of dot-product self-attention to $\ell_2$ perturbations can actually be controlled by several key quantities associated with the attention weight matrices and the unperturbed input. We empirically validate our theoretical findings through several examples, offering new insights for achieving low sensitivity in dot-product self-attention against $\ell_2$ input perturbations.

## 1 INTRODUCTION

The self-attention mechanism (Bahdanau et al., 2014; Vaswani et al., 2017) has become a major building block in many modern deep learning-based systems, achieving state-of-the-art performance in various applications such as vision and natural language processing. In particular, dot-product self-attention (Vaswani et al., 2017) is one of the most popular architectures used by many best-performing networks such as the well-known Transformer architecture and its variants, and enabled successful applications such as large language models (LLM) (Brown et al., 2020; Bubeck et al., 2023) and vision transformers (ViT) (Dosovitskiy et al., 2021; Radford et al., 2021).

Unlike traditional neural network building blocks such as convolutional layers, whose structure and behavior are well understood, the self-attention mechanism has more involved mathematical properties. For example, for a simple convolutional layer, it is well known that its operator norm is bounded (Sedghi et al., 2019), and convolution is a Lipschitz operation that always produces bounded outputs given bounded inputs (Delattre et al., 2023). However, for the popular dot-product self-attention mechanism, existing work has shown that they are surprisingly, not (globally) Lipschitz (Kim et al., 2021). The lack of Lipschitzness indicates that dot-product self-attention can theoretically be very sensitive to its input, which can impede stable learning (Qi et al., 2023) and lead to poor robustness (Zhou et al., 2022; Cisse et al., 2017). Although several architectures have been proposed to amend the popular dot-product attention mechanism to achieve Lipschitzness and bounded sensitivity (Kim et al., 2021; Dasoulas et al., 2021; Qi et al., 2023), none of them are popular in large-scale networks deployed in production, and it is still an open challenge to understand why the non-Lipschitz dot-product attention mechanism can work well in practice.

In this work, instead of amending the network structure to achieve bounded sensitivity, we aim to analyze the *local* sensitivity of the *unmodified* dot-product attention mechanism directly. Despite being non-Lipschitz, we derived the first non-vacuous bound for local sensitivity of unmodified self-attention mechanism, using mathematical techniques from optimization and matrix theory. Our result consists of a theorem deciphering a few key quantities associated with the sensitivity of the dot-product self-attention operation, related to the attention weight matrix and their inputs. The new result gives us insights on controlling the local sensitivity of a Transformer. In particular, we found that the local sensitivity of the self-attention layer is directly related to the norm of its input, thus

theoretically explaining the necessity of using layer normalization (Ba et al., 2016) in the popular Transformer architecture (Xiong et al., 2020). In addition, it allows us to utilize the recent progress of 1-Lipschitz feedforward neural network layers, such as orthogonal layers (Trockman & Kolter, 2021; Prach & Lampert, 2022) and the SDP-based Lipschitz Layer (SLL) (Araujo et al., 2023), to control the local sensitivity of Transformers. Note that since the self-attention layer is non-Lipschitz, naively applying 1-Lipschitz layers could not provide any guarantees without our new local results.

We confirm our theoretical findings on a few practical vision transformers by quantifying their local sensitivity against $\ell_2$ norm input perturbations. Our experiments show that our derived local sensitivity bounds are practical for vision Transformers and significantly improve against a naive approach for sensitivity analysis. In addition, we also use gradient ascent to find the maximum sensitivity empirically, and demonstrate that our theoretical bounds and empirical measurements are well-aligned. By varying the design parameters of the vision transformers (*e.g.*, number of attention heads and number of tokens), our theory predicts the observed changes in local sensitivity. As a by-product of our bounds, we can give non-trial adversarial robustness guarantees for vision transformers with standard dot-product self-attention mechanisms. The main contributions of this work are:

- We are the first to consider a fine-grained theoretical analysis of *local sensitivity* bounds of *unmodified* dot-product self-attention mechanism, contributing to the mathematical understanding of this popular network structure. Despite the non-Lipschitzness of dot-product self-attention, our local bounds are non-trivial and *non-vacuous* when validated on practical vision Transformers.
- Our results give great insights into achieving low sensitivity on dot-product self-attention-based Transformers. It enables us to borrow the recently developed algebraic tricks on training globally 1-Lipschitz feedforward networks to provably improve the *local* sensitivity of Transformers.
- Our theoretical results are validated through the empirical evaluation of a large range of Transformers trained with different design parameters. In addition, our tight bounds allow us to achieve non-trivial deterministic robustness guarantees for vision Transformers without modifying the dot-product self-attention mechanism.

We introduce the problem formulation in Sec. 3, present our main results in Sec. 4, and a comprehensive set of experiments to validate our analysis in Sec. 5. We defer all proofs to the Appendix.

## 2 RELATED WORK

**Lipschitz and Regularity of Self-Attention.** Since the first Lispchitz analysis of dot-product self-attention by Kim et al. (2021), which showed that the standard dot-product self-attention is not Lipschitz, a large number of works have tried to propose variants of the original dot-product self-attention to enforce this property (Kim et al., 2021; Qi et al., 2023; Fei et al., 2022; Dasoulas et al., 2021; Ye et al., 2023). For example, Qi et al. (2023) proposed scaled cosine similarity attention instead of dot product attention and demonstrated the Lispchitz properties of this new layer. Another type of work (Vuckovic et al., 2021) has studied the regularity of attention under a mathematical framework that uses measure theory and integral operators to model attention. Under this new framework, they show that the attention mechanism is regular (under some specific assumptions) with respect to the 1-Wasserstein distance. While this work generalizes the work of Kim et al. (2021), the regularity over the 1-Wasserstein distance is not commonly used in practice.

**Neural networks with prescribed Lipschitz Constant.** Recently, researchers have designed neural networks with prescribed Lipschitz constant in order to better control the stability (Miyato et al., 2018), robustness (Zhang et al., 2021; Prach & Lampert, 2022; Meunier et al., 2022; Zhang et al., 2022; Araujo et al., 2023; Wang & Manchester, 2023; Li et al., 2019; Trockman & Kolter, 2021; Singla & Feizi, 2021; Yu et al., 2022; Xu et al., 2022), and generalization (Bartlett et al., 2017) of the network. However, most of these technique comes with important design choices with respect to the architecture that are not common in networks with state-of-the-art performance.

**Robustness of Transformer Networks.** Since dot-product self-attention is not (globally) Lipschitz, robustness cannot be derived from Lipschitz continuity (Tsuzuku et al., 2018). Existing work used randomized smoothing (Cohen et al., 2019) to probabilistically certify their robustness (Carlini et al., 2023; Wu et al., 2023). Randomized smoothing suffers from high computational cost and the inherent probabilistic nature of the certificate. In this work, our bounds can be used to provide non-trivial and *deterministic* certified accuracy for Transformers with unmodified dot-product self-attention.

## 3 Preliminaries and Problem Formulation

**Notation and Background** We denote the spectral norm and the Frobenius norm as $\|\cdot\|$ and $\|\cdot\|_F$, respectively. Two useful facts are $\|AB\|_F \leq \|A\|\|B\|_F$, and $\|A\| = \|A^\mathsf{T}\|$. Given any two matrices $A$ and $B$, their Kronecker product of $A$ and $B$ is denoted as $A \otimes B$. We denote the vectorization operation as vec. Let $e_i$ denote an $n$-dimensional vector whose $i$-th entry is 1 and all other entries are 0. The $n \times n$ identity matrix is denoted by $I_n$. The standard softmax mapping on matrices is denoted as softmax. We know that softmax is 1-Lipschitz, i.e. $\|\mathrm{softmax}(A) - \mathrm{softmax}(B)\|_F \leq \|A - B\|_F$ for any two matrices $A$ and $B$ that have the same dimension.

**Dot-Product Self-Attention** Let $x_1, x_2, \ldots, x_n$ be a sequence of $n$ vectors, where $x_i \in \mathbb{R}^d$. For vision tasks, each $x_i$ is a patch. This sequence is represented as a matrix $X$. The dot-product multi-head self-attention maps $\mathbb{R}^{n \times d}$ to $\mathbb{R}^{n \times d}$. With $h$ heads, the $l$-th head maps $\mathbb{R}^{n \times d}$ to $\mathbb{R}^{n \times d/h}$ as:

$$X = \begin{bmatrix} - & x_1^\mathsf{T} & - \\ & \vdots & \\ - & x_n^\mathsf{T} & - \end{bmatrix} \in \mathbb{R}^{n \times d} \qquad Y_l = \mathrm{softmax}\left( \frac{XW_l^Q (XW_l^K)^\mathsf{T}}{\sqrt{d/h}} \right) XW_l^V$$

where $W_l^Q, W_l^K, W_l^V \in \mathbb{R}^{d \times d/h}$ denote the weight matrices for the $l$-th head, and the softmax operation is applied in a row-wise manner. Finally, the outputs of all heads are concatenated as

$$f(X) = [Y_1, \ldots, Y_h] W^O = \sum_{l=1}^{h} Y_l W_l^O,$$

where $W^O = [(W_1^O)^\mathsf{T}, \ldots, (W_h^O)^\mathsf{T}]^\mathsf{T} \in \mathbb{R}^{d/h \times d}$ gives the weight for the linear combination of the outputs from all the heads. For simplicity, we denote the notation $P_l(X)$, and the dot-product self-attention can be rewritten as:

$$P_l(X) = \mathrm{softmax}\left( \frac{XW_l^Q (XW_l^K)^\mathsf{T}}{\sqrt{d/h}} \right) \quad (1) \qquad f(X) = \sum_{l=1}^{h} P_l(X) XW_l^V W_l^O \qquad (2)$$

**Residual structure.** Dot-product self-attention is typically used in a residual form. In this case, the output is defined as $f(X) = X + \sum_{l=1}^{h} P_l(X) XW_l^V W_l^O$.

**Problem Statement.** It is well-known that (2) is not globally Lipschitz (Kim et al., 2021). We are interested in analyzing the local sensitivity of dot-product self-attention. We consider the following model which unifies (2) and its residual variant with $H \in \mathbb{R}^{n \times n}$:

$$F(X) = HX + \sum_{l=1}^{h} P_l(X) XW_l^V W_l^O. \qquad (3)$$

If $H = 0$, then (3) recovers the standard dot-product self-attention (2). If $H = I$, then (3) reduces to the residual setting. Given a local input point $X$ and some small positive scalar $\epsilon$, we want to prove a bound in the following form:

$$\|F(X') - F(X)\|_F \leq \delta(X, \epsilon) \quad \text{for } X' \text{ satisfying} \quad \|X' - X\|_F \leq \epsilon \qquad (4)$$

where $F$ is defined by (3). In principle, the tightest choice of $\delta(X, \epsilon)$ is given by the solution to the following constrained optimization problem

$$\max_{X': \|X' - X\|_F \leq \epsilon} \|F(X') - F(X)\|_F \qquad (5)$$

One can use the projected gradient ascent method to search solutions for (5). However, there are no polynomial-time guarantees in solving the above problem globally. In addition, the bound (5) does not bring any insights for how to control the local sensitivity via network structure design. The goal of this paper is to develop a spectrum of choices for $\delta(X, \epsilon)$ that can capture the trade-off between tightness, tractability, and interpretability.

Once we figure out an efficient way to compute $\delta(X, \epsilon)$ for the above problem, we can immediately apply the analysis in a recursive manner to solve the local sensitivity analysis of multi-layer networks consisting of various dot-product self-attention layers. Specifically, consider a $N$-layer network:

$$F(X) = f^N \circ f^{N-1} \circ \cdots \circ f^0(X) \qquad (6)$$

where $f^k$ is either a dot-product self-attention layer (3) or a globally 1-Lipschitz operation. Applying the local sensitivity analysis in a recursive manner, we will be able to compute $\delta(X, \epsilon)$ for bounding the end-to-end local sensitivity of (6) as described by (4). Such a bound can be used to prove the robustness of $F$ on the data point $X$ subject to adversarially chosen $\ell_2$ perturbations. For example, the following result connects the local bound $\delta(X, \epsilon)$ to certified robustness.

**Proposition 1.** *Suppose $F$ is a classifier that maps any input $X$ to the output as defined by* (6). *The $j$-th entry of $F(X)$ is denoted as $[F(X)]_j$, which gives the logits value for the $j$-th label class. The predicted label for $X$ is given by $\arg\max_j[F(X)]_j$. Given an input $X$ with the true label $y$ satisfying $y = \arg\max_j[F(x)]_j$, if we have*

$$\mathcal{M}_{\mathbf{f}}(X) := [F(X)]_y - \max_{j \neq y}[F(X)]_j > \sqrt{2}\delta(X, \epsilon),$$

*then for every $\tau$ satisfying $\|\tau\|_F \leq \epsilon$, we must have $\arg\max_j[F(X + \tau)]_j = y$.*

The proof for the above result is almost identical to (Tsuzuku et al., 2018, Proposition 1), hence omitted. The above proposition provides a way to compute the certified robust accuracy of dot-product self-attention using our local sensitivity analysis. We emphasize that the analysis of $\delta(X, \epsilon)$ is not the same as obtaining a local Lipschitz bound. The difference is clarified in the appendix.

## 4 LOCAL ANALYSIS OF DOT-PRODUCT SELF-ATTENTION

In this section, we perform the local sensitivity analysis for the case where $F$ is defined by (3). Specifically, we have $F(X) = HX + \sum_{l=1}^h P_l(X)XW_l^V W_l^O$ for either $H = 0$ or $H = I$. First, the following bound based on the splitting trick is standard:

$$\|F(X') - F(X)\|_F$$

$$=\left\|H(X' - X) + \sum_{l=1}^h P_l(X')X'W_l^V W_l^O - \sum_{l=1}^h P_l(X)XW_l^V W_l^O\right\|_F$$

$$=\left\|H(X' - X) + \sum_{l=1}^h P_l(X)(X' - X)W_l^V W_l^O + \sum_{l=1}^h (P_l(X') - P_l(X))X'W_l^V W_l^O\right\|_F$$

$$\leq\left\|H(X' - X) + \sum_{l=1}^h P_l(X)(X' - X)W_l^V W_l^O\right\|_F + \left\|\sum_{l=1}^h (P_l(X') - P_l(X))X'W_l^V W_l^O\right\|_F$$

Next, we will bound the two terms on the right side. We use the following notation

$$\Delta_1 = \left\|H(X' - X) + \sum_{l=1}^h P_l(X)(X' - X)W_l^V W_l^O\right\|_F \tag{7}$$

$$\Delta_2 = \left\|\sum_{l=1}^h (P_l(X') - P_l(X))X'W_l^V W_l^O\right\|_F \tag{8}$$

If we can derive bounds in the form of $\Delta_1 \leq \delta_1(X, \epsilon)$ and $\Delta_2 \leq \delta_2(X, \epsilon)$, then we immediately have the bound $\|F(X') - F(X)\|_F \leq \delta(X, \epsilon) := \delta_1(X, \epsilon) + \delta_2(X, \epsilon)$. Our analysis addresses how to reduce the conservatism in deriving $\delta_1(X, \epsilon)$ and $\delta_2(X, \epsilon)$.

**Reducing conservatism in deriving $\delta_1(X, \epsilon)$** One may naively bound $\Delta_1$ using the property of matrix norms as follows (see the appendix for a detailed derivation):

$$\Delta_1 \leq \left(\|H\| + \sum_{l=1}^h \|W_l^V W_l^O\|\|P_l(X)\|\right)\epsilon. \tag{9}$$

The above bound is informative in showing that one can potentially control $\Delta_1$ by constraining the spectral norm of $\{W_l^V, W_l^O\}_{l=1}^h$. However, the above bound can be too loose quantitatively. In contrast, the best possible bound for $\Delta_1$ can be obtained as the solution to the following problem:

$$\max_{X':\|X'-X\|_F \leq \epsilon} \left\|H(X' - X) + \sum_{l=1}^h P_l(X)(X' - X)W_l^V W_l^O\right\|_F \tag{10}$$

The above problem actually has an analytical solution. This leads to our first result stated as follows.

**Lemma 1** (Key Sensitivity Metric). *The exact solution to the optimization problem* (10) *is given by*

$$\max_{X':\|X'-X\|_F\le\epsilon}\Big\|H(X'-X)+\sum_{l=1}^{h}P_l(X)(X'-X)W_l^V W_l^O\Big\|_F = \zeta(X)\epsilon, \tag{11}$$

*where $\zeta(X)$ is defined as*

$$\zeta(X) = \Big\|H\otimes I_n + \sum_{l=1}^{h}(P_l(X)\otimes(W_l^V W_l^O)^\mathsf{T})\Big\| \tag{12}$$

*Consequently, we have $\Delta_1\le\delta_1(X,\epsilon)=\zeta(X)\epsilon$, for all $X'$ satisfying $\|X'-X\|_F\le\epsilon$.*

A detailed proof for Lemma 1 is presented in the appendix. The main analysis idea is based on the following key identity:

$$\mathrm{vec}\left(\sum_{l=1}^{h}(P_l(X)(X'-X)W_l^V W_l^O)^\mathsf{T}\right)=\left(\sum_{l=1}^{h}(P_l(X)\otimes(W_l^V W_l^O)^\mathsf{T})\right)\mathrm{vec}((X'-X)^\mathsf{T})$$

which enables us to solve (10) exactly via viewing it as a largest singular value problem. The quantity $\zeta(X)$ is termed as the key sensitivity metric which quantifies the local sensitivity of the self-attention around the data point $X$ due to the error $\Delta_1$. The computation of this metric is reasonably scalable so that one can efficiently compute this metric for ViT used for CIFAR10. Later, we will show that this term is the dominating term in the bound $\Delta_1+\Delta_2$, and hence one should calculate this term exactly when fine-grained sensitivity analysis is needed.

**Reducing Conservatism in Deriving $\delta_2(X,\epsilon)$** The best possible bound for $\Delta_2$ is the solution to the following constrained maximization problem:

$$\max_{X':\|X'-X\|_F\le\epsilon}\Big\|\sum_{l=1}^{h}(P_l(X')-P_l(X))X'W_l^V W_l^O\Big\|_F \tag{13}$$

One can apply the projected gradient ascent method to the above problem. However, there are no guarantees that the resultant solution is global due to the form of the cost function. The solution from the gradient ascent method only provides lower bound for (13). To obtain a more tractable upper bound, it is straightforward to apply the triangle inequality to show that (13) can be bounded by the following term:

$$\sum_{l=1}^{h}\left(\max_{X':\|X'-X\|_F\le\epsilon}\big\|(P_l(X')-P_l(X))\big\|_F\cdot\max_{X':\|X'-X\|_F\le\epsilon}\|X'W_l^V W_l^O\|\right), \tag{14}$$

which involve two maximization problems. Now we discuss these two problems separately.

To address the term $\max_{X':\|X'-X\|_F\le\epsilon}\|X'W_l^V W_l^O\|$, we can apply the triangle inequality and obtain the following tractable upper bound:

$$\max_{X':\|X'-X\|_F\le\epsilon}\|X'W_l^V W_l^O\|\le\|XW_l^V W_l^O\|+\|W_l^V W_l^O\|\epsilon \tag{15}$$

The above upper bound can be efficiently calculated via power iteration, and is less conservative than the naive bound $\|W_l^V W_l^O\|(\|X\|+\epsilon)$. Later, we will show that the above upper bound is reasonable for the purpose of upper bounding $\Delta_1+\Delta_2$, since replacing it with the lower bounds obtained by the projected gradient ascent method does not affect the final overall bound value significantly.

Next, we discuss how to address

$$\max_{X':\|X'-X\|_F\le\epsilon}\big\|(P_l(X')-P_l(X))\big\|_F \tag{16}$$

Again, one can apply the projected gradient ascent method to search for lower bounds for the above quantity. We are more interested in obtaining less conservative upper bounds that are computationally tracable. Since $\mathrm{softmax}$ is 1-Lipschitz, we can show the following holds for any $X$:

$$\|P_l(X')-P_l(X)\|_F\le\frac{1}{\sqrt{d/h}}\|X'W_l^Q(W_l^K)^\mathsf{T}(X')^\mathsf{T}-XW_l^Q(W_l^K)^\mathsf{T}X^\mathsf{T}\|_F \tag{17}$$

Denoting $\Gamma = X' - X$. If $\|X' - X\|_F \leq \epsilon$, then we have $\|\Gamma\|_F \leq \epsilon$. We immediately have

$$\|P_l(X') - P_l(X)\|_F \leq \frac{1}{\sqrt{d/h}} \|(X + \Gamma)W_l^Q(W_l^K)^\mathsf{T}(X + \Gamma)^\mathsf{T} - XW_l^Q(W_l^K)^\mathsf{T}X^\mathsf{T}\|_F$$

$$= \frac{1}{\sqrt{d/h}} \|\Gamma W_l^Q(W_l^K)^\mathsf{T}X^\mathsf{T} + XW_l^Q(W_l^K)^\mathsf{T}\Gamma^\mathsf{T} + \Gamma W_l^Q(W_l^K)^\mathsf{T}\Gamma^\mathsf{T}\|_F$$

which leads to the following bound for (16):

$$\frac{1}{\sqrt{d/h}} \max_{\Gamma: \|\Gamma\|_F \leq \epsilon} \|\Gamma W_l^Q(W_l^K)^\mathsf{T}X^\mathsf{T} + XW_l^Q(W_l^K)^\mathsf{T}\Gamma^\mathsf{T} + \Gamma W_l^Q(W_l^K)^\mathsf{T}\Gamma^\mathsf{T}\|_F \qquad (18)$$

The above problem can be searched using the projected ascent method. However, there are no polynomial-time guarantees in maximizing a fourth-order polynomial subject to a quadratic norm constraint. Fortunately, when $\epsilon$ is reasonably small, the following bound is not loose due to the negligible effects of the higher-order term. We can obtain the following bound:

$$\max_{\Gamma: \|\Gamma\|_F \leq \epsilon} \frac{1}{\sqrt{d/h}} \|\Gamma W_l^Q(W_l^K)^\mathsf{T}X^\mathsf{T} + XW_l^Q(W_l^K)^\mathsf{T}\Gamma^\mathsf{T}\|_F + \max_{\Gamma: \|\Gamma\|_F \leq \epsilon} \frac{1}{\sqrt{d/h}} \|\Gamma W_l^Q(W_l^K)^\mathsf{T}\Gamma^\mathsf{T}\|_F$$

We can easily bound the second term as

$$\max_{\Gamma: \|\Gamma\|_F \leq \epsilon} \frac{1}{\sqrt{d/h}} \|\Gamma W_l^Q(W_l^K)^\mathsf{T}\Gamma^\mathsf{T}\|_F \leq \frac{\epsilon^2}{\sqrt{d/h}} \|W_l^Q(W_l^K)^\mathsf{T}\|. \qquad (19)$$

In addition, the exact value of the first term can be calculated using the following lemma.

**Lemma 2.** *The following relation holds*

$$\max_{\Gamma: \|\Gamma\|_F \leq \epsilon} \frac{1}{\sqrt{d/h}} \|\Gamma W_l^Q(W_l^K)^\mathsf{T}X^\mathsf{T} + XW_l^Q(W_l^K)^\mathsf{T}\Gamma^\mathsf{T}\|_F = \frac{1}{\sqrt{d/h}} \|M_l(X)\|\epsilon,$$

*where $M_l(X)$ is given by the following specific matrix*

$$M_l(X) = I_n \otimes \begin{bmatrix} x_1^\mathsf{T} W_l^K (W_l^Q)^\mathsf{T} \\ \vdots \\ x_n^\mathsf{T} W_l^K (W_l^Q)^\mathsf{T} \end{bmatrix} + \sum_{i=1}^{n} (e_i \otimes I_n) \otimes (x_i^\mathsf{T} W_l^Q (W_l^K)^\mathsf{T}). \qquad (20)$$

The dimension of $M_l(X)$ can be quite high. A bound that can be quickly computed is given by

$$\|M_l(X)\| \leq \xi_l(X) := \left( \|W_l^Q(W_l^K)^\mathsf{T}X^\mathsf{T}\| + \|XW_l^Q(W_l^K)^\mathsf{T}\| \right). \qquad (21)$$

Putting all the bounds that we have obtained, we can get the following local sensitivity result.

**Theorem 1.** *Consider the dot-product self-attention model* (3). *Suppose an input point $X$ is given. For any $X'$ satisfying $\|X' - X\|_F \leq \epsilon$, we have*

$$\|F(X') - F(X)\|_F \leq \epsilon \left( \zeta(X) + \frac{1}{\sqrt{d/h}} \sum_{l=1}^{h} (\|M_l(X)\| + \|W_l^Q(W_l^K)^\mathsf{T}\|\epsilon)(\|XW_l^V W_l^O\| + \|W_l^V W_l^O\|\epsilon) \right)$$

$$\leq \epsilon \left( \zeta(X) + \frac{1}{\sqrt{d/h}} \sum_{l=1}^{h} (\xi_l(X) + \|W_l^Q(W_l^K)^\mathsf{T}\|\epsilon)(\|XW_l^V W_l^O\| + \|W_l^V W_l^O\|\epsilon) \right),$$

*where $\zeta(X)$ is given by (12), $M_l(X)$ is given by (20), and $\xi_l(X)$ is defined by (21).*

The above bounds can be used to obtain the first non-trivial certified robustness result of dot-product self-attention on CIFAR10. We will show this in the numerical result section.

**Insights for Network Design.** Our theory should not suggest that weight matrices and data with small magnitude are necessarily better for network design. The right interpretation is that our bound can be used to quantify the robustness/performance trade-off for dot-product self-attention and achieve non-vacuous certified robust accuracy. For example, if we make $\|W_l^Q\|$, $\|W_l^K\|$, $\|W_l^V\|$, $\|W_l^O\|$, and $\|X\|$ small, then our local sensitivity bound is guaranteed to be small. However, too much regularity may limit expressive power. From this insight, it is possible to borrow the recent advancements on how to constraining weight norms from the Lipschitz network literature to design dot-product self-attention layers with weight norm being controlled. In addition, the insight on the need of controlling $\|X\|$ further justifies the use of layer normalization in training such attention layers.

## 5 EXPERIMENTS

In this section, we will study the conservatism introduced in our fine-grained analysis and how to use these local bounds in a scalable manner. Furthermore, we will study how our analysis can be used to inform the design of robust self-attention blocks when applied to ViT on the CIFAR-10 dataset and explore the trade-offs between performance and robustness of our regularized ViT. To the best of our knowledge, we present the first non-trivial $\ell_2$-certified robust accuracy result for ViT using standard dot-product self-attention with deterministic certificates.

### 5.1 STUDYING CONSERVATISM IN THE LOCAL BOUND

In our fine-grained local sensitivity analysis of multi-head self-attention, each step used to upper-bound the output introduces conservatism. Of course, these steps are important for making the local upper-bound computationally practical and scalable. We aim to show that the conservatism introduced by these choices does not significantly degrade the effectiveness of our estimate and that our key sensitivity metric of Lemma 1 is quite informative for quantifying the robustness for small values of the $\ell_2$ input perturbation level $\epsilon$. Recall that for the multi-head self-attention block $F$ ($H = I$ for the standard residual attention block), our analysis considers two major terms, $\Delta_1$ and $\Delta_2$, in upper-bounding the local perturbed output at an input $X$ with respect to the Frobenius norm.

$$\|F(X') - F(X)\|_F \leq$$

$$\underbrace{\left\|(X' - X) + \sum_{l=1}^{h} P_l(X)(X' - X)W_l^V W_l^O\right\|_F}_{\Delta_1 \leq \delta_1(X,\epsilon)} + \underbrace{\left\|\sum_{l=1}^{h} (P_l(X') - P_l(X))X'W_l^V W_l^O\right\|_F}_{\Delta_2 \leq \delta_2(X,\epsilon)}$$

For the first term $\Delta_1$, we have already established a tight upper-bound $\delta_1(X, \epsilon) = \zeta(X)\epsilon$ using the key sensitivity metric, which can be readily computed by power-iterations. Therefore the majority of the conservatism will be introduced when upper-bounding $\Delta_2 \leq \delta_2(X, \epsilon)$. As discussed in section 4, the best possible bound for $\Delta_2$ directly maximizes over $X'$ as in Eq (13), but has no guaranteed global solution. For that reason, we further estimate and seek upper-bounds to the sum of separate inner-maximization problems as in Eq (14). Since each term is isolated in the sum, its enough to consider the single-head attention case to determine the effect of $\Delta_2$.

**Single-head Case: Bounding $\Delta_2$**  To upper bound $\Delta_2$ in the single-head case, we need to compute bounds for the following two multiplicative terms.

$$\Delta_2 \leq \underbrace{\left\|P(X') - P(X)\right\|_F}_{\Delta_{2,1} \leq \delta_{2,1}(X,\epsilon)} \cdot \underbrace{\left\|X'W^V W^O\right\|}_{\Delta_{2,2} \leq \delta_{2,2}(X,\epsilon)}$$

For bounding $\Delta_{2,1}$, our fine-grained analysis of Theorem 1 offers us the following two upper bounds from our fine-grained analysis,

$$\delta_{2,1}^{(1)} = \frac{\epsilon}{\sqrt{d/h}} \left(\|M(X)\| + \epsilon\|W^Q(W^K)^\mathsf{T}\|\right)$$

$$\delta_{2,1}^{(2)} = \frac{\epsilon}{\sqrt{d/h}} \left(\|W^Q(W_l^K)^\mathsf{T} X^\mathsf{T}\| + \|XW^Q(W^K)^\mathsf{T}\| + \epsilon\|W^Q(W^K)^\mathsf{T}\|\right)$$

where $M(X)$ is given in (20). Clearly $\delta_{2,2}^{(1)}$ is tighter than $\delta_{2,1}^{(2)}$, but more expensive to compute since $M(X)$ is a $(n^2 \times nd)$ matrix. For $\Delta_{2,2}$, we consider the bound $\delta_{2,2}^{(1)}$ from Theorem 1 and compare it to a more conservative naive bound $\delta_{2,2}^{(2)}$ that was considered in our analysis.

$$\delta_{2,2}^{(1)}(X,\epsilon) = \left(\|XW^V W^O\| + \epsilon\|W^V W^O\|\right), \quad \delta_{2,2}^{(2)}(X,\epsilon) = \|W^V W^O\|(\|X\| + \epsilon) \qquad (22)$$

To get an idea of the conservatism introduced by each proposed bound, we compare the bounds at a point $X$ over a range of $\epsilon$ values in $[0, 0.1]$ (X is normalized as $\|X\|_F = 4$, the number of patches. Later we will apply unit projection to each patch before each attention unit). In order to better understand the tightness of our bounds, we also compare these bounds against their respective lower-bound given by directly performing PGD on $\max_{X',\|X'-X\|_F \leq \epsilon} \Delta_{2,1}$ and $\max_{X',\|X'-X\|_F \leq \epsilon} \Delta_{2,2}$.

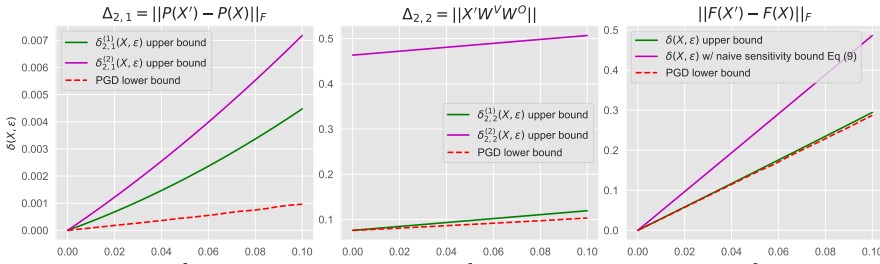

Figure 1: We compare the the proposed single-head bounds for $\Delta_1$, $\Delta_2$ and the end-to-end multi-head attention bound $\|F(X') - F(X)\|_F$ across the input perturbation bound $\epsilon$. The PGD lower bound is given by directly maximizing $\max_{X', \|X'-X\|_F \leq \epsilon} \Delta_{2,1}$ and $\max_{X', \|X'-X\|_F \leq \epsilon} \Delta_{2,2}$.

These results can be seen in Figure 1. It is most important to note that for $\Delta_{2,1}$, although there seems to be a significant gap in the PGD lower bound and our upper bounds, the relative scale of these terms is small compared to the input perturbation for small values of $\epsilon$ (the bounds are quadratic in $\epsilon$). Also for this reason, there may not be much improvement from using the tighter bound $\delta_{2,1}^{(1)}$ in most cases.

For $\Delta_{2,2}$, our bound $\delta_{2,2}(X, \epsilon)$ is fairly tight in this range of $\epsilon$, noting that this value is further multiplied with $\delta_{2,1}$. This renders the term $\Delta_2$ negligible for small values of $\epsilon$ and a controlled spectral norm of the input $\|X\|$. From this study, it is more clear that $\Delta_1$ and the key sensitivity metric Lemma 1 meaningfully measure sensitivity in a local regime. In this sense, the conservatism introduced to estimate $\Delta_2$ is justified.

**Multi-head Case: End-to-End Tightness** To further validate that $\Delta_1$ captures the majority of the sensitivity for controlled inputs, we compare our upper bound to the PGD lower bound of the entire multi-head attention layer with $h = 8$ heads. Through our previous study, summing over the heads, we can justify the following practical upper bound which coincides with Theorem 1 (with $H = 0$).

$$\|F(X') - F(X)\|_F \leq \delta(X, \epsilon) = \zeta(X)\epsilon + \sum_{l=1}^{h} \delta_{2,1,l}^{(2)}(X, \epsilon) \cdot \delta_{2,2,l}^{(1)}(X, \epsilon) \tag{23}$$

This multi-head bound is also evaluated in Figure 1, alongside the single-head components. To emphasize how crucial our key sensitivity metric is for tightness, we also compare against the same bound (23), but using the naive sensitivity bound from Eq (9). As a lower bound, we compare against PGD which directly maximizes $\max_{X', \|X'-X\| \leq \epsilon} \|F(X') - F(X)\|_F$. It becomes clear that when the spectral norm of the input $X$ is controlled and $\epsilon$ is small, our upper bound is tight. That is because the contribution of $\Delta_2$ is small and our bound on $\Delta_1$ is tight. However, when the input norm of $X$ is large, the conservative terms of $\delta_{2,1}(X, \epsilon)$, which depend on $X$ begin to drive the estimate upwards and loosen our bound. We will use these insights to design a more robust ViT to achieve non-trivial certified accuracy on the classification task CIFAR10.

## 5.2 APPLICATIONS TO CERTIFIED ACCURACY ON CIFAR10

In this section, we will now apply our local upper bound of the dot-product attention unit to achieve an end-to-end local bound of ViT. With this local bound and the margin argument given in Proposition 1, we can achieve certified robust accuracy to $\ell_2$-bounded attacks. Informed by our upper bound, we can make fine-grained design choices of ViT that trade-off performance and robustness.

**Controlling Local Sensitivity of Vision Transformer.** Although our theory supports the standards dot-product self-attention unit commonly used in ViT, the bound needs to be propagated through other modules such as feed-forward layers and layer normalization. We use a standard ViT architecture which uses residual attention and residual feed-forward blocks. Based on our local upper bound, we can make important changes to enable greater robustness in our upper bound.

**Layer Projection.** Modules such as `LayerNorm` are not globally Lipschitz, but can be crucial to the performance of ViT. We instead replace `LayerNorm` units with `LayerProject` defined by

$$\texttt{LayerProject}(x, R) = \begin{cases} x/\|x\|_2 \cdot R & \text{if} \quad \|x\|_2 > R, \\ x & \text{otherwise} \end{cases}, \tag{24}$$

Figure 2: Certified robust accuracy on CIFAR10 using our local sensitivity bounds under many combinations of ViT architecture parameters (number of layers, heads, and norm of weight matrices).

where $R = \sqrt{d}$ is set to mimic the behavior of `LayerNorm`. Because projection to a closed convex set is 1-Lipschitz, we can seamlessly propagate our upper bound and maintain the desired input scale. Additionally, `LayerProject` with $R = 1$, is applied before each attention head so that the spectral norm of the entire input $\|X\|$ is approximately 1 and so our upper-bound remains controlled at each attention unit.

**Lipschitz Constrained Layers.** The upper bound of Theorem 1, especially the key sensitivity metric, will depend directly on the spectral norm of the attention weight matrices $(W_l^Q, W_l^K, W_l^V, W_l^O)$. In order to keep the expansion of our upper-bound through each layer low, we constrain the norms of these weights through SDP-based Lipschitz Layer (SLL) Araujo et al. (2023). We also constrain all feed-forward modules and patch embedding units to be 1-Lipshitz to easily propagate our upper bound with unit expansion through ViT. This is not an uncommon choice of parameterization, as orthogonal-ViT Fei et al. (2022) has been proposed, using orthogonal 1-Lipschitz layers in the attention unit to improve generalization on smaller data sets.

$\ell_2$ **Certified Robust Accuracy on CIFAR10.** We will now apply our end-to-end local upper bound to obtain certified robust accuracy on the vision task CIFAR10. We study the effect of different ViT architecture parameters such as the number of attention heads, number of layers, image patch size, and Lipschitz constant constraint of the attention weights.

This result on certified robust accuracy presented in Figure 2 is, to our knowledge, the first non-trivial $\ell_2$-certified robustness result using standard dot-product attention, with non-zero robustness up to $\epsilon \approx 36/255$. The results also show that many of these architectural factors will introduce a trade-off between clean accuracy and certified robustness. This is to be expected since, for example, the number of layers will cause our upper bound to compound, but it is also crucial for clean performance.

In addition to 1-Lipschitz layers, we found it crucial to additionally constrain the Lipschitz constant of the weights $W^V$ and $W^O$ to be a fraction of 1, which directly affects the key sensitivity metric, the largest contributing factor of expansion in our local bound on ViT. It's also important to note that without directly exploiting the residual structure in the key sensitivity metric of Lemma 1 (instead of the naive bound) or using Lipschitz controlled weights, no meaningful certified robust accuracy could be achieved.

## 6 CONCLUSION

This work has provided a comprehensive examination of the standard dot-product self-attention mechanism, a critical component in many machine learning models, especially in the context of vision transformers. Despite the well-established understanding that dot-product self-attention is not globally Lipschitz, this study has delved into a fine-grained local sensitivity analysis, shedding light on its behavior when subjected to input feature perturbations. The theoretical results presented in this paper have been empirically validated through a comprehensive set of experiments. These findings provide a deeper understanding of how to mitigate sensitivity issues in dot product self-attention when faced with $\ell_2$ input perturbations.

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

# A LOCAL SENSITIVITY ANALYSIS VS. LOCAL LIPSCHITZ ANALYSIS

A local Lipschitz analysis aims at showing that for any two points $(X', X'')$ in the $\epsilon$-ball around $X$, the following bound holds

$$\|F(X') - F(X'')\|_F \leq L\|X' - X''\|_F.$$

This is stronger than our sensitivity analysis, and may be too strong for establishing non-trivial certified robustness results of dot-product self-attention. If one can show the above local Lipschitz bound holds, then clearly one can choose $\delta(X, \epsilon) = L\epsilon$. However, given our local sensitivity bound (4), one cannot guarantee local Lipschitzness. Specifically, the local Lipschitz bound allows choosing arbitrary two points in the $\epsilon$-neighborhood of the original input $X$. In contrast, our local sensitivity analysis is in a weak sense that the bound can only tell us the derivations of $F(X')$ from $F(X)$.

# B DETAILED DERIVATIONS OF OUR BOUNDS

## B.1 PROOF OF PROPOSITION 1

Let $X$ be an input and suppose that the margin of the classifier $F$ at $X$ satisfies $\mathcal{M}_{\mathbf{f}}(X) > \sqrt{2}\delta(X, \epsilon)$. Then for any $||\tau||_2 \leq \epsilon$ we have:

$$
\begin{aligned}
\mathcal{M}_{\mathbf{f}}(X + \tau) =& [F(X + \tau)]_y - \max_{j \neq y}[F(X + \tau)]_j \\
=& [F(X)]_y - \max_{j \neq y}[F(X)]_j - ([F(X)]_y - [F(X + \tau)]_y) + (\max_{j \neq y}[F(X)]_j - \max_{j \neq y}[F(X + \tau)]_j) \\
=& [F(X)]_y - \max_{j \neq y}[F(X)]_k - \begin{bmatrix} 1 \\ -1 \end{bmatrix}^\top \begin{bmatrix} [F(X)]_y - [F(X + \tau)]_y \\ \max_{j \neq y}[F(X)]_j - \max_{j \neq y}[F(X + \tau)]_j \end{bmatrix} \\
\geq& [F(X)]_y - \max_{j \neq y}[F(X)]_j - \left| \begin{bmatrix} 1 \\ -1 \end{bmatrix}^\top \begin{bmatrix} [F(X)]_y - [F(X + \tau)]_y \\ \max_{j \neq y}[F(X)]_j - \max_{j \neq y}[F(X + \tau)]_j \end{bmatrix} \right| \\
\geq& [F(X)]_y - \max_{j \neq y}[F(X)]_j - \left\| \begin{bmatrix} 1 \\ -1 \end{bmatrix} \right\|_2 \|F(X) - F(X + \tau)\|_2 \\
\geq& \mathcal{M}_{\mathbf{f}}(X) - \sqrt{2}\delta(X, \epsilon) > 0
\end{aligned}
$$

Therefore, $\arg\max_j[F(X + \tau)]_j = y$ for all $\tau$ such that $\|\tau\|_2 \leq \epsilon$. This completes the proof.

## B.2 A DETAILED DERIVATION OF (9)

By the triangle inequality, we have

$$
\begin{aligned}
\Delta_1 &\leq \|H\| + \sum_{l=1}^{h}\|P_l(X)(X' - X)W_l^V W_l^O\|_F \\
&\leq \|H\| + \sum_{l=1}^{h}\|P_l(X)\|\|(X' - X)W_l^V W_l^O\|_F \\
&\leq \|H\| + \sum_{l=1}^{h}\|P_l(X)\|\|X' - X\|_F\|W_l^V W_l^O\|,
\end{aligned}
$$

which gives the stated bound.

### B.3 PROOF OF LEMMA 1

Obviously, we have

$$\left\|H(X'-X)+\sum_{l=1}^{h}P_l(X)(X'-X)W_l^V W_l^O\right\|_F$$

$$=\left\|(X'-X)^\mathsf{T}H^\mathsf{T}+\sum_{l=1}^{h}(W_l^V W_l^O)^\mathsf{T}(X'-X)^\mathsf{T}(P_l(X))^\mathsf{T}\right\|_F$$

Since $(A\otimes B)\operatorname{vec}(V)=\operatorname{vec}(BVA^\mathsf{T})$, we must have

$$\operatorname{vec}\left((X'-X)^\mathsf{T}H^\mathsf{T}+\sum_{l=1}^{h}(W_l^V W_l^O)^\mathsf{T}(X'-X)^\mathsf{T}(P_l(X))^\mathsf{T}\right)$$

$$=\left((H\otimes I_n)+\sum_{l=1}^{h}P_l(X)\otimes(W_l^V W_l^O)^\mathsf{T}\right)\operatorname{vec}((X'-X)^\mathsf{T})$$

Therefore, we are minimizing the $\ell_2$ norm of the right side of the above equation subject to an $\ell_2$ norm constraint on $\operatorname{vec}((X'-X)^\mathsf{T})$. Therefore, the maximum value is achieved by the product of the largest singular value of $\left((H\otimes I_n)+\sum_{l=1}^{h}P_l(X)\otimes(W_l^V W_l^O)^\mathsf{T}\right)$ and $\epsilon$.

### B.4 PROOF OF LEMMA 2

To prove this lemma, we denote $\Gamma_i = x_i' - x_i \in \mathbb{R}^d$. Set $\beta_{ij} = \Gamma_i^\mathsf{T}W^Q(W^K)^\mathsf{T}x_j + x_i^\mathsf{T}W^Q(W^K)^\mathsf{T}\Gamma_j$. We can augment $\{\beta_{ij}\}$ as the following big vector:

$$\Lambda=\begin{bmatrix}\beta_{11}\\\beta_{12}\\\vdots\\\beta_{1n}\\\beta_{21}\\\beta_{22}\\\vdots\\\beta_{n1}\\\vdots\\\beta_{nn}\end{bmatrix}=M(X)\begin{bmatrix}\Gamma_1\\\Gamma_2\\\vdots\\\Gamma_n\end{bmatrix}$$

where $M(X)$ is given by the following specific matrix

$$M(x)$$

$$=\begin{bmatrix}x_1^\mathsf{T}(W^K(W^Q)^\mathsf{T}+W^Q(W^K)^\mathsf{T}) & 0 & 0 & \cdots & 0\\x_2^\mathsf{T}W^K(W^Q)^\mathsf{T} & x_1^\mathsf{T}W^Q(W^K)^\mathsf{T} & 0 & \cdots & 0\\x_3^\mathsf{T}W^K(W^Q)^\mathsf{T} & 0 & x_1^\mathsf{T}W^Q(W^K)^\mathsf{T} & \cdots & 0\\\vdots & \vdots & \vdots & \vdots & \vdots\\x_n^\mathsf{T}W^K(W^Q)^\mathsf{T} & 0 & 0 & \cdots & x_1^\mathsf{T}W^Q(W^K)^\mathsf{T}\\\vdots & \vdots & \vdots & \vdots & \vdots\end{bmatrix}$$

$$=I_n\otimes\begin{bmatrix}x_1^\mathsf{T}W^K(W^Q)^\mathsf{T}\\\vdots\\x_n^\mathsf{T}W^K(W^Q)^\mathsf{T}\end{bmatrix}+\sum_{i=1}^{n}(e_i\otimes I_n)\otimes(x_i^\mathsf{T}W^Q(W^K)^\mathsf{T}).$$

Based on the above largest singular value interpretation, we can obtain the desired conclusion.

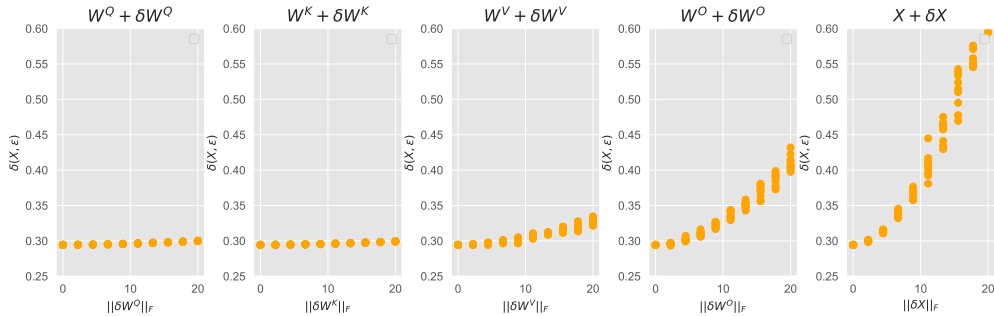

Figure 3: We perform an ablation on the weight and input and its effect on the derived local upper-bound of Theorem 1. Perturbations are applied to a trained set of self-attention weights and input $(W^Q, W^K, W^V, W^O, X)$, perturbing one element of the tuple at a time. A total of $10$ samples are taken for each parameter and each weight/input perturbation size. We set $\varepsilon = 0.1$ for all samples.

### B.5 PROOF OF THEOREM 1

We can combine Lemma 1, the bound (14), the bound (14), the bound (19), and Lemma 2 together, and the resultant bound is the desired one stated in this theorem.

## C ADDITIONAL EXPERIMENTS

### C.1 ABLATION STUDY OF ATTENTION WEIGHTS

In this section, we study more closely the effects of each parameter in the multi-head attention map on our bound in Theorem 1. To do this, we consider the weights $(W^Q, W^K, W^V, W^O)$ from the first layer of a ViT trained on CIFAR10 and a normalized input $X$ (as the input undergoes projection prior to each attention layer in our architecture). We then perturb each element while keeping the others fixed, observing how our upper-bound (1) is affected with increasing parameter perturbation size. For the experiment, we fix $\varepsilon = 0.1$. A total of $10$ samples are taken for each weight and each perturbation size. The study in found in Figure 3. Based on this study, we can observe that the weights $W^V$, $W^O$ and $X$ account for much of the sensitivity of our bound, therefore, controlling the norm of these weights and the input is crucial to control our bound.

### C.2 TIGHTNESS OF BOUND AND INPUT NORM

In order to further study the affect of the input norm size and how it affects the tightness of our bound, we perform an extended study similar to the one in Figure 1 for several input norm scales. In this study, we are looking at the first residual self-attention layer of a pretrained ViT with $8$ heads and evaluating all proposed bounds discussed in section 5.1. We find that the bounds suggested in our paper remain tight as long as the input norm is not too large. For large inputs values, our bound eventually loses some effectiveness. This further justifies why we should perform pre-layer projection if one desires to maintain non-trivial robustness using our proposed bound.

### C.3 APPLICATIONS TO SENTIMENT ANALYSIS: WORD EMBEDDING ROBUSTNESS

In order to broaden the application domains of our theory, we apply our local sensitivity bounds to a sentiment analysis task, where Transformer architectures are commonly utilized Kenton & Toutanova (2019). In this section, we provide a robustness study on the Stanford Sentiment Tree-bank (SST) dataset Socher et al. (2013) using transformers based on the BERT architecture Kenton & Toutanova (2019). We are using the version of SST that classifies sentences into two classes which indicate a positive or negative sentiment. As in previous works Wang et al. (2020); Zhu et al. (2019); Li & Qiu (2020); Xu et al. (2023), we reason about $\ell_2$ bounded adversarially perturbations on the word embedding space, as it is not easy to formulate perturbations on the tokens themselves

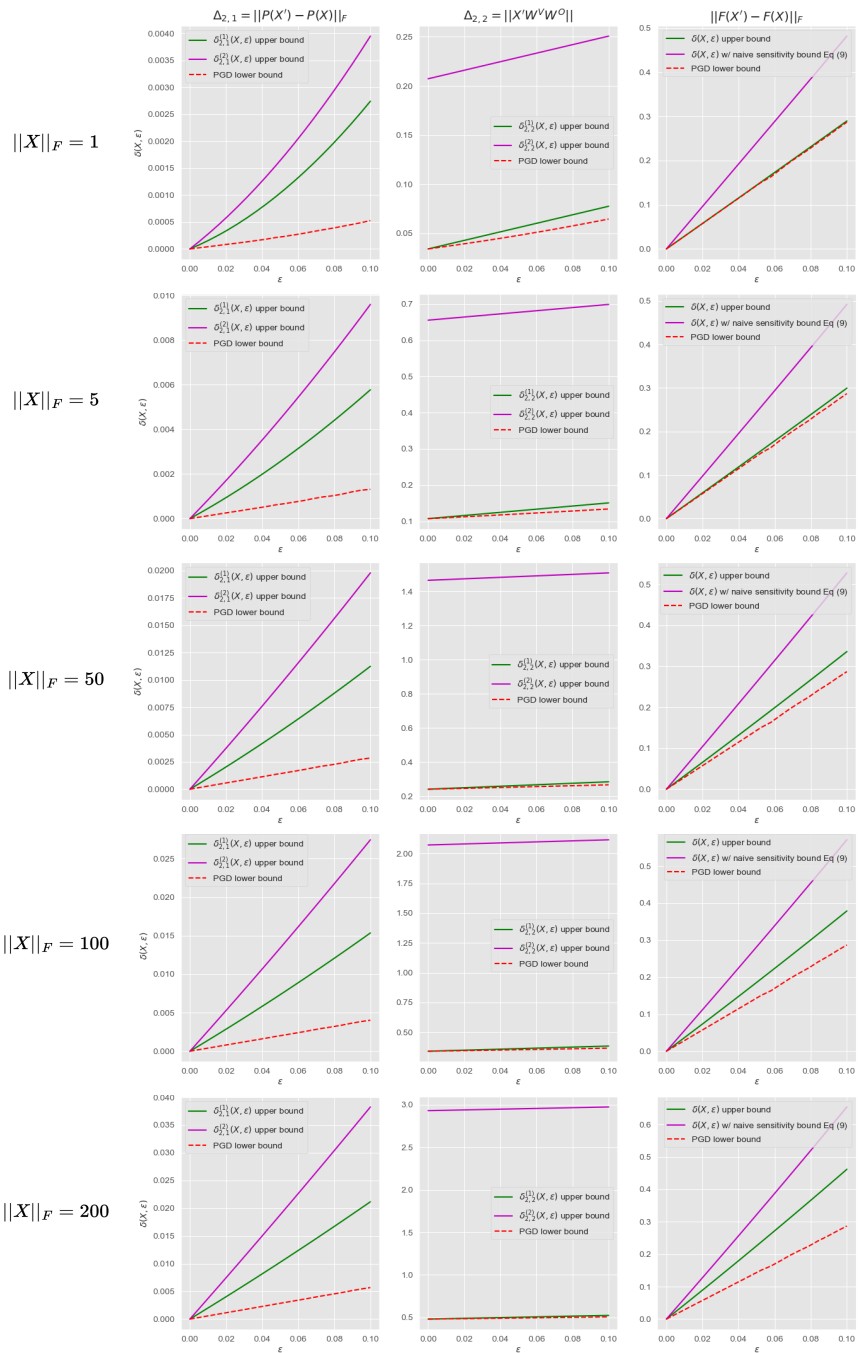

Figure 4: We repeat the conservatism study of our bound in Fig 1 with different input norms, to study the how this affects the tightness of our bound for a single multi-head attention layer.

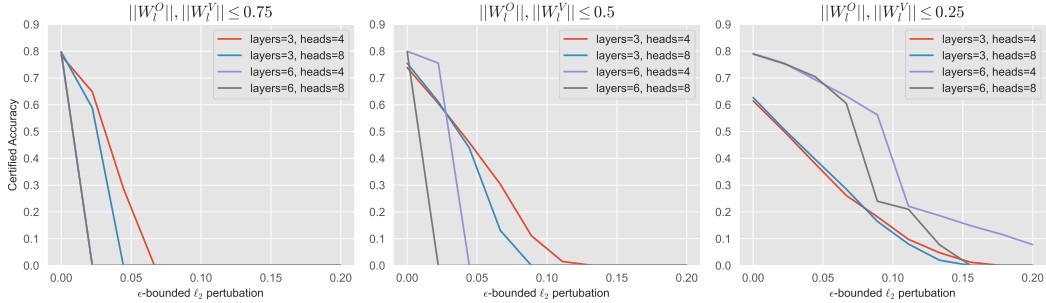

Figure 5: Certified robust accuracy on the SST using our local sensitivity bounds under many combinations of small BERT architecture parameters (number of layers, heads, and norm of weight matrices).

using $\ell_2$ perturbations. We must point out that this is a common limitation of applying sensitivity analysis to NLP benchmarks, as already noted in other works Hou et al. (2022). The certified robustness radii we obtained measured in the $\ell_2$ norm are similar to those in prior work *without* using dot-product attention Xu et al. (2023). The results show that our sensitivity analysis bounds are indeed non-vacuous.

**Experiment Setup for SST Sentiment Data-set**   Similarly to Section 5.2, we will examine the certified accuracy of several architectures and choices of weight norm restrictions. As mentioned before, we consider perturbations applied directly to word embeddings. In this case, $\epsilon$ describes the radius of the raw perturbation, rather a ratio of the pixel value in our vision task. The self-attention architecture designs are identical to the ones used for ViT, except we consider a embedding dimension of $d = 64$ and 32 tokens per input (i.e. $X \in \mathbb{R}^{32 \times 64}$). We consider combinations of self-attention units with layers in $\{3, 6\}$ and number of heads in $\{4, 8\}$. Additionally, we train each architecture constraining the output attention weights $W^V, W^O$ to have spectral norm in $\{0.25, 0.5, 0.75\}$ using the same SLL layer. The results are presented below in Figure 5. In this case, we see that adding regularity does not necessarily decrease clean accuracy because the task is rather simple. By controlling the bound sufficiently, we can even sustain good robust accuracy while applying more layers (see the right-most panel in Figure 5).

