# OpenReview forum: "Fine-grained Local Sensitivity Analysis of Standard Dot-Product Self-Attention"
_ICLR.cc/2024/Conference — Submitted to ICLR 2024_

### Official Review · Reviewer_BhUy · 2023-10-29

**Soundness:** 3 good
**Presentation:** 3 good
**Contribution:** 2 fair
**Rating:** 5
**Confidence:** 2

**Summary:**

This paper studies the local sensitivity of dot-product self-attention in Transformers. Though the outputs of all heads is not globally Lipchitz, a weaker condition, i.e., the local sensitivity can be theoretically analyzed by providing a upper bound. Besides, the upper bound is empirically verified and certification on practical models is also given.

**Strengths:**

- Upper bound of local sensitivity analyse is derived
- Numerical validations are also provided to support the fact that, the upper bound is tight and reasonable

**Weaknesses:**

- The dimension of some matrices are undefined, e.g., $W^O \in R^{d \times d}$ and $H \in R^{n \times n}$?
- Solving Eq. (10) requires SVD for the n-by-d matrix. How to ensure the computational efficiency?
- To bound the second term in Eq. (14), the author uses the triangle inequality to obtain the upper bound at first. However, this can be also obtained with a closed-form solution? This is because the objective function and constraint are both linear.

**Questions:**

- Before Proposition 1, the authors mention the robustness under l_2 perturbation. How about using l_\inf perturbations for robustness when compared to the adversarially chosen l_2 perturbations? In this case, Eq. (5) will be changed to the l_\inf norm but I’m not sure the used techniques are still valid.

---

> ### Author Response · Authors · 2023-11-19
> **Response to Reviewer BhUy**
>
> Thanks for your comments. We believe that the significance of our contribution has been underestimated. We share Reviewer d9ub's opinion that our paper has made significant contributions in achieving **the first non-zero certified robustness result of standard dot-product self-attention networks on CIFAR**. We further address all your individual comments as below.
>
>
> > The dimension of some matrices are undefined, e.g., $W^O$ and $H$?
>
> We consider a standard setting with $W^O\in \mathbb{R}^{(d/h) \times d}$ and $H\in \mathbb{R}^{n\times n}$. We have now made the dimensions of these matrices clear in the revised paper
>
>
> >Solving Eq. (10) requires SVD for the n-by-d matrix. How to ensure the computational efficiency?
>
> To be clear, our approach only requires computing the spectral norm, and SVD is not needed for solving the spectral norm. In our paper, we just use the power iteration method which is known to be very efficient in computing the spectral norm and has been used many times in deep learning (e.g. Tsuzuku'18,  Miyato'18, Meunier'22).
>
>
>
> > To bound the second term in Eq. (14), the author uses the triangle inequality to obtain the upper bound at first. However, this can be also obtained with a closed-form solution? This is because the objective function and constraint are both linear.
>
> Unfortunately, there is no closed-form solution for the second term of Eq. (14), since this is a problem maximizing the spectral norm under a quadratic constraint. The objective function $||X' W^V W^O||$ is not a linear function of $X'$ due to the appearance of the spectral norm. The constraint $||X'-X||_F \leq \epsilon$ is also not linear in $X'$ due to the appearance of the Frobenius norm. It is actually quadratic.
> For problems minimizing the spectral norm under a quadratic constraint, it is possible to reformulate them as semidefinite programs (SDPs). However, the second term of Eq. (14) requires maximizing the spectral norm, and hence it is difficult to obtain a bound tighter than our current bound based on triangle inequality.
>
>
> > Before Proposition 1, the authors mention the robustness under $\ell_2$ perturbation. How about using $\ell_\infty$ perturbations for robustness when compared to the adversarially chosen $\ell_2$ perturbations? In this case, Eq. (5) will be changed to the $\ell_\infty$ norm but I’m not sure the used techniques are still valid.
>
> Extending our fine-grained $\ell_2$ analysis to the $\ell_\infty$ case is definitely non-trivial and would require some major changes, since we are using the fact that softmax is $1$-Lipschitz with respect to the $\ell_2$ norm. We want to emphasize that it is totally reasonable for our current paper to focus on the $\ell_2$ perturbation cases, and there are many papers published in top machine learning venues (e.g Singla'21, Trockman'21, Meunier'22, Singla'22, Prach'22, Araujo'23, Wang'23,Hu'23) that only focus on robustness and sensitivity of neural networks under the $\ell_2$ setting. Our paper is the first obtaining a non-vacuous certified robustness result of standard dot-product self-attention networks on CIFAR.
>
>
> [Singla'21]. Sahil Singla and Soheil Feizi. Skew orthogonal convolutions. ICML
>
> [Trockman'21] Asher Trockman and J Zico Kolter. Orthogonalizing convolutional layers with the Cayley transform. ICLR
>
> [Meunier'22] Laurent Meunier, Blaise J Delattre, Alexandre Araujo, and Alexandre Allauzen. A dynamical system
> perspective for lipschitz neural networks. ICML
>
> [Singla'22] Sahil Singla, Surbhi Singla, and Soheil Feizi. Improved deterministic l2 robustness on CIFAR-10 and CIFAR-100. ICLR
>
> [Prach'22] Bernd Prach and Christoph H Lampert. Almost-orthogonal layers for efficient general-purpose lipschitz networks. ECCV
>
> [Araujo'23] Alexandre Araujo, Aaron J Havens, Blaise Delattre, Alexandre Allauzen, and Bin Hu. A unified algebraic perspective on lipschitz neural networks. ICLR
>
> [Wang'23] Ruigang Wang and Ian Manchester. Direct parameterization of lipschitz-bounded deep networks. ICML
>
> [Hu'23] Kai Hu, Andy Zou, Zifan Wang, Klas Leino, and Matt Fredrikson. Unlocking deterministic robustness certification on imagenet. NeurIPS

---

> > ### Comment · Reviewer_BhUy · 2023-11-23
> >
> > thanks for the authors' response.
> >
> > If the author(s) would like to focus on **the first non-zero certified robustness result of standard dot-product self-attention networks on CIFAR**, there are some references that are missing:
> >
> > https://openreview.net/forum?id=BJxwPJHFwS
> >
> > Shi, Z., Zhang, H., Chang, K.W., Huang, M. and Hsieh, C.J., 2020. Robustness verification for transformers. ICLR 2020.
> >
> > Bonaert, G., Dimitrov, D.I., Baader, M. and Vechev, M., 2021, June. Fast and precise certification of transformers. In Proceedings of the 42nd ACM SIGPLAN International Conference on Programming Language Design and Implementation (pp. 466-481).
> >
> > ---
> >
> > Sorry for the late reply but my score will remain unchanged.

---

### Official Review · Reviewer_VymQ · 2023-10-31

**Soundness:** 2 fair
**Presentation:** 2 fair
**Contribution:** 2 fair
**Rating:** 5
**Confidence:** 4

**Summary:**

This paper aim to theoretically analyze the sensitivity of the self attention mechanism. Local perturbations are imposed on the weights, and authors quantify the relationship between the sensitivity between the input, weight matrices, etc. Experiments are done to validate the theory, and insights are provided to achieve more stable self attention structure.

**Strengths:**

1. This paper captures a common problem of the popular Transformer model: self attention mechanism can be sensitive. The work quantifies the sensitivity and provides insight into how to make the self attention structure stable. This topic is important in the performance of Transformer model, which is widely applied in NLP, CV tasks.
2. I do not have doubt on the theoretical results, as they are clearly derived.
3. The experiments are closely related with the theory.

**Weaknesses:**

1. My main concern is that this work does not provide enough contribution. In Section 4, the gap caused by perturbation is derived. However, these results are not novel, in fact, they are easy to derive. The main idea of Section 4 is just finding a Lipschitz constant to bound the gap when perturbation is added to input. This can be easily done if we take derivative over input X and find an upper bound for the Frobenius norm of the gradient over X. In some other works, the closed form gradients (maybe over $W^Q,W^K$, but similar to gradient over X) are easily derived, e.g, Tian, Yuandong, et al. "Scan and Snap: Understanding Training Dynamics and Token Composition in 1-layer Transformer." arXiv preprint arXiv:2305.16380 (2023). Thus, I do not think the theory has much contribution.
2. The theory in Section 4 implies that weight matrices and data with small magnitude is better. However, 'small magnitude' does not mean a self attention mechanism is a good model. Consider an extreme case where all weight matrices are close to 0, then the attention mechanism has poor representation ability. We usually require a model with both expressivity and stability, while in this work, the expressivity is ignored.

**Questions:**

1. How to theoretically guarantee that a model can both have good expressivity and stability?
2. When weights $W^Q,W^K,W^V$ follows some specific distribution, can the sensitivity bound be improved? Or the bound is only related to the magnitude of weights?

---

> ### Author Response · Authors · 2023-11-19
> **Response to Reviewer VymQ**
>
> Thanks for your comments. We believe that our contribution has been misunderstood. We share Reviewer d9ub's opinion that our paper has made significant contributions in achieving **non-vacuous certified robustness result of standard dot-product self-attention networks on CIFAR**. We further address all your individual comments as below.
>
> > However, these results are not novel, in fact, they are easy to derive. This can be easily done if we take derivative over input X and find an upper bound for the Frobenius norm of the gradient over X.
>
> We think our results are novel for the following reason. Our local sensitivity analysis is not a local Lipschitz bound result (see the explanation in Appendix A of our paper), and **our analysis is really needed to achieve our main goal that is to produce tight local bounds which can lead to non-vacuous certified robustness of standard dot-product self-attention networks on realistic data sets such as CIFAR.** We emphasize that getting **non-vacuous** certified robust accuracy is an important topic that is being studied in the deep learning field (e.g. see Singla'21, Trockman'21, Meunier'22, Singla'22, Prach'22, Araujo'23, Wang'23,Hu'23), and this is very different from the cases when one studies generalization bounds which are typically used to provide design guidelines and hence not required to be non-vacuous. If one follows the suggestions in this comment (e.g. finding upper bound for the Frobenius norm of the gradient), then the resultant bounds (e.g. the local Lipschitz bound in a concurrent submission available at https://openreview.net/forum?id=mivL0akE5E is exactly derived using this gradient norm idea) is too loose and can only produce 0 certified robust accuracy for self-attention networks on CIFAR.  Now we elaborate as follows.
>
> - Non-vacuous certified robustness: Given a data point $X$, a classifier $F$ is said to be certifiably robust at radius $\epsilon > 0$ at this data point with label $y$ if for all $\tau$ such that $||\tau|| \leq \epsilon$, we have $arg max_j [F (X+\delta)]_j = y$. Given a perturbation level $\epsilon$, the certified robust accuracy is defined to be the percentage of the data points (in the testing test) where the classifier $F$ is certifiably robust at radius $\epsilon$, and this notion of certified robustness has been adopted in many recent works (e.g. Singla'21, Trockman'21, Meunier'22, Singla'22, Prach'22, Araujo'23, Wang'23,Hu'23). Based on our Proposition 1, our local sensitivity bound can be combined with the prediction margin to calculate the certified robust accuracy. As commented by Reviewer d9ub, achieving non-vacuous certified accuracy for dot-product self-attention on CIFAR10 task is one of our main contributions.  If ones tries to develop a local Lipschitz bound and uses it for certified robustness, the bound is usually too loose and the robust accuracy will be 0. For example, the local Lipschitz bound in a concurrent submission (Specformer) available at https://openreview.net/forum?id=mivL0akE5E is exactly derived using this gradient norm idea (see Theorem 4.3 of that paper). If we compare Theorem 4.3 of the SpecFormer paper with our bound (see table below), we can see our bounds are better by magnitude. The consequence is that Theorem 4.3 of the SpecFormer paper can only achieve 0 certified accuracy on CIFAR. **Our key sensitivity metric (Lemma 1 in our paper) is novel and crucial for obtaining a non-vacuous certified accuracy result.**
>
> | $\ell_2$ perturbation bound $\epsilon$ | .01| .02|.03|.04|.05|.06| .07 |.08 | .09 | .10 |
> |----|---|--------|--------|--------|--------|---------|---------|---------|---------|---------|
> | PGD Lower Bound | 0.0286 | 0.0573 | 0.0860 | 0.1147 | 0.1427 | 0.1719  | 0.2008  | 0.2295  | 0.2582  | 0.2865  |
> | Local Bound Theorem 1. (ours) | 0.0291 | 0.0582 | 0.0875 | 0.1168 | 0.1462 | 0.1757  | 0.2052  | 0.2348  | 0.2646| 0.2943|
> | SpecFormer Local Lipschitz Bound (Gradient-based)| 16.901 | 34.403 | 52.515 | 71.245 | 90.600 | 110.589 | 131.219 | 152.499 | 174.436 | 197.036 |
>
> [Figure Comparing to the Gradient-based Bound](https://drive.google.com/file/d/1riLDjkjXMbf0bmi3Ux6t0puZ_z60mSFq/view?usp=sharing)
>
>
> - Our analysis is not a local Lipschitz constant analysis: To reduce conservatism, the local bounds derived in our paper are actually quite different from a local Lipschitz constant bound. Our local bound $||X-X'||\leq \epsilon \implies ||F(X)-F(X')|| \leq \delta(X, \epsilon)$ fixes $X$ and only allows $X'$ to vary. This is sufficient for calculating certified robust accuracy. The Lipschitz constant bound you describe $||F(X'')-F(X')||\leq L(X, \epsilon)||X''-X'||$ actually holds for any two points $(X',X'')$ in the $\epsilon$-ball around $X$.  This introduces unnecessary conservatism when applied to compute the certified robustness.

---

> ### Author Response · Authors · 2023-11-19
> **Response to Reviewer VymQ**
>
> > The theory in Section 4 implies that weight matrices and data with small magnitude is better. In this work, the expressivity is ignored.
>
> Our theory should not imply that weight matrices and data with small magnitude are necessarily better for network design. Previous study on certifiably robust networks (e.g Singla'21, Trockman'21, Meunier'22, Prach'22, Araujo'23, Wang'23) has revealed that there is a trade-off between deterministic $\ell_2$ certified robustness and the clean performance for standard feed-forward networks or residual networks (dot-product self-attention has not been covered in these previous works). The right interpretation is that our bound can be used to quantify such a robustness/performance trade-off for dot-product self-attention (based on our bound, one can sacrifice the clean performance to achieve non-vacuous certified robust accuracy for standard dot-product self-attention). Importantly, as constraining the weight/data norm can improve the certified robustness, there is a price to pay, i.e.  the clean accuracy will drop.  So we never claim that one should always maximize the robustness. Instead, our theory offers the tool to quantify the robustness/performance trade-off for dot-product self-attention such that one can explore this trade-off for various tasks at hand (different tasks require different levels of robustness). We have also revised our paper accordingly to better reflect the above point.
>
> > How to theoretically guarantee that a model can both have good expressivity and stability?
>
> As mentioned before, our focus is more on certified robustness rather than stability (which is typically used for generalization). Now we address the above comment via emphasizing the trade-off between certified robustness and clean performance. Our results and other papers on certifiably robust networks (e.g Singla'21, Trockman'21, Meunier'22, Prach'22, Araujo'23, Wang'23) all demonstrate that there is a trade-off between deterministic $\ell_2$ certified robustness and the clean performance (expressivity). Currently, improving certified robustness typically leads to degraded performance. One just navigates the optimal Pareto trade-off between robustness and expressivity depending on the possible level of perturbations for the tasks at hand.  Continuing to boost the clean performance of certifiably robust models is an ongoing research effort in the robust learning community. Our main contribution is on characterizing the robustness/performance trade-off for dot-product self-attention, which is a very reasonable self-contained topic. How to improve the structures of transformers to obtain better Pareto trade-off curves for robustness/robustness is beyond the scope of our paper, and should be investigated in the future. Our work serves as a foundation for such future study.
>
> > When weights follows some specific distribution, can the sensitivity bound be improved?
>
> We follow the standard setup in the certified robustness literature  and consider the certified robustness of a trained model with deterministic weight (Singla'21, Trockman'21, Meunier'22, Prach'22, Araujo'23, Wang'23). Evaluating the robust accuracy for a fixed trained model makes sense, since robustness certification is typically applied to a trained model. To the best of our knowledge, certified robustness under distributions of the network weight has not been extensively studied in the robust learning field. It is unclear how to extend our analysis or any other existing deterministic $\ell_2$ certified robustness analysis to such a setting considering a weight distribution.
>
> >Or the bound is only related to the magnitude of weights?
>
> Our bound in Theorem 1 not only depends on the spectral norm of the weights, but is highly dependent on the actual matrix elements and how they interact with the input and other attention heads. This is especially true in the "key sensitivity metric" of eq. (12) since we are looking at the spectral norm after summing many attention weights together, making our bound more fine-grained and less conservative.
> Here's a simple example: Suppose we have an attention layer with two attention heads. We set $H=0$, $W^Q_1 = W^Q_2$, $W^K_1=W^K_2$ ($P(X)_1=P(X)_2$), and $W^V_1 = W^V_2$. Then if $W^O_2 = -1/2*W^O_1$, the sensitivity metric of Eq (12) is: $||\sum_i P(X)_i \otimes (W^V_i W^O_i)^T|| = 1/2 ||P(X)_1 \otimes (W^V_1W^O_1)^T||$. However, if $W^O_2= 1/2 W^O_1$, then we have $ 3/2 ||P(X)_1 \otimes (W^V_1W^O_1)^T||$ which is a bound 3 times larger. In this case, we have not changed the spectral norm of any weight matrix, only the sign direction. But our key sensitivity metric changes significantly. To summarize, our bound is mainly related to our key sensitivity metric (Lemma 1).  Decreasing the spectral norms of $(W^Q, W^K, W^V, W^O)$ can eventually lead to a decreased value of the key sensitivity metric. However, that is not the only way to decrease the key sensitivity metric.

---

> ### Author Response · Authors · 2023-11-19
> **The list of the references mentioned in our response**
>
> For your convenience, here we list the detailed information of the references mentioned in our above response. Hopefully this makes reading our response easier. We are also willing to address any follow-up questions.
>
> [Singla'21]. Sahil Singla and Soheil Feizi. Skew orthogonal convolutions. ICML
>
> [Trockman'21] Asher Trockman and J Zico Kolter. Orthogonalizing convolutional layers with the Cayley transform. ICLR
>
> [Meunier'22] Laurent Meunier, Blaise J Delattre, Alexandre Araujo, and Alexandre Allauzen. A dynamical system perspective for lipschitz neural networks. ICML
>
> [Singla'22] Sahil Singla, Surbhi Singla, and Soheil Feizi. Improved deterministic l2 robustness on CIFAR-10 and CIFAR-100. ICLR
>
> [Prach'22] Bernd Prach and Christoph H Lampert. Almost-orthogonal layers for efficient general-purpose lipschitz networks. ECCV
>
> [Araujo'23] Alexandre Araujo, Aaron J Havens, Blaise Delattre, Alexandre Allauzen, and Bin Hu. A unified algebraic perspective on lipschitz neural networks. ICLR
>
> [Wang'23] Ruigang Wang and Ian Manchester. Direct parameterization of lipschitz-bounded deep networks. ICML
>
> [Hu'23] Kai Hu, Andy Zou, Zifan Wang, Klas Leino, and Matt Fredrikson. Unlocking deterministic robustness certification on imagenet. NeurIPS

---

> ### Comment · Reviewer_VymQ · 2023-11-23
> **Reply to authors**
>
> Thanks for clarification. I have changed the score. But I still think the work is a little below the margin considering the theoretical contribution is not sufficient.

---

### Official Review · Reviewer_d9ub · 2023-11-10

**Soundness:** 3 good
**Presentation:** 3 good
**Contribution:** 3 good
**Rating:** 8
**Confidence:** 3

**Summary:**

This paper provides a fine-grained theoretical analysis on the local sensitivity of self-attention. The primary constrained optimization for this local sensitivity is $\max_{X': ||X' - X||_F \leq \epsilon} || F(X') - F(X) ||_F$ where $F(X)$ is the residual self attention. The authors divide $|| F(X') - F(X) ||_F$ into $\Delta_1$ and $\Delta_2$ (equation 7 & 8). For $\Delta_1$, the authors provide an analytical upper bound (**first contribution** of this paper) as

$$\xi(x) = || H \bigotimes I_n + \sum_{l = 1}^h (P_l(X) \bigotimes (W_l^V W_l^O)^\top) ||$$

For $\Delta_2$, the authors first apply triangle inequality to divide equation 13 into the perturbation on self-attention score matrix (equation 17) and $|| X' W_l^V W_l^O||$ (equation 15). **The second contribution** of this paper is on bounding equation 17 as Lemma 2 and equation 19. Putting them all together, we obtain an upper bound for $|| F(X') - F(X) ||_F$.

In the experiments, the authors first study the $\Delta_1$ and $\Delta_2$ values versus the PGD low bound across epsilon values in single and multi (8) head cases. The authors also analyze ViT's certified robust accuracy on CIFAR10 task, and provide a nonzero robustness $\epsilon \sim 36/255$ (**third contribution**).

**Strengths:**

The proof organization of this paper is pretty clear and easy to follow in section 4. The authors meticulously described the looseness of each naive bound and strategies to further tighten the bound.

This local sensitivity analysis would be insightful for both adversarial and general machine learning community.

The experiments are also conducted on real-world tasks (ViT on CIFAR10), which makes this theoretical analysis practical on understanding the robustness of self-attention.

**Weaknesses:**

There are multiple naive bounds described in theory but not evaluated in practice. For example, equation 9 for bounding $\Delta_1$ and $||W_l^V W_l^O||(||X|| + \epsilon)$ for bounding $||X' W_l^V W_l^O||$ should also be evaluated in Figure 1 to make the conservatism argument strong.


Minor:

typo in equation 5: $||F(X') - F(x)||$ should be $||F(X') - F(X)||$

**Questions:**

Is it possible to perform another trial of robustness experiments on NLP tasks (text classification, entailment, etc.)? The analysis of this paper is applied to general self-attention and it is definitely great to see a practical evaluation on vision tasks. But it would be even better to see if the same robustness argument is applicable across domains.

In the network design section, it is mentioned that Theorem 1 would shed light on constraining weight norms for self-attention. It would be nice to see a concrete use case. For example, given a particular quadruple $(W_Q, W_K, W_V, W_O)$ and an input $X$, could we ablate on each weight individually and use the Theorem 1 to predict the local sensitivity?

Overall, this is a good paper, but I believe the evaluation section could be further improved. I would give a weak accept score at this moment, but I am willing to raise my score if my above concerns / questions are addressed.

---

> ### Author Response · Authors · 2023-11-19
> **Response to Reviewer d9ub**
>
> Thank you for your valuable feedback. We have addressed your comments below.
>
> > There are multiple naive bounds described in theory but not evaluated in practice... they
> should also be evaluated in Figure 1 to make the conservatism argument strong.
>
> We agree that evaluating these naive bounds, especially the naive estimate of eq (9), would make the conservatism argument strong.  We have revised the experimental section (Section 5) and Figure 1 with the additional naive bounds. The results further show that our choice of the key-sensitivity metric is crucial for getting tight local sensitivity bounds of dot-product self-attention. We have also included an additional study on the tightness of our bound similar to Figure 1 for several input norm values found in Appendix C.2, Figure 4. We hope this provides a more complete picture of how our bound and architecture deal with conservatism.
>
> >Is it possible to perform another trial of robustness experiments on NLP tasks (text classification, entailment, etc.)? The analysis of this paper is applied to general self-attention and it is definitely great to see a practical evaluation on vision tasks. But it would be even better to see if the same robustness argument is applicable across domains.
>
> Thanks for this comment. We followed your instruction and produced a robustness study of dot-product self-attention on the Stanford Sentiment Tree-bank (SST) dataset. We also revised our paper to include these new results in Appendix C.3 and Figure 5. We follow the setup in Xu’23, Wang’20, Zhu’20, and Li’21, and consider the deterministic $\ell_2$ certified robustness on the word embedding space (we use the BERT embedding from Devlin'19). Our analysis here only serves as a proof of concept for using our bounds to NLP tasks, and we acknowledge that considering perturbations on the word embedding space (instead of the tokens directly) is a common limitation of applying sensitivity analysis to NLP benchmarks, as shared in many other papers (Hou’23). The certified robustness radii that we obtained is measured using $\ell_2$ norm, and this is similar to those in prior work *without* using dot-product attention (Xu’23). Our results demonstrate that our sensitivity analysis bounds lead to non-vacuous robustness results on SST. For more details, see Appendix C of our revised paper. If the reviewer further wants us to work on other specific tasks, please let us know and we are willing to provide more results.
>
> [Xu’23] Xu et al. Certifiably Robust Transformers with 1-Lipschitz Self-Attention.
> [Hou’23] Hou et al. TextGrad: Advancing Robustness Evaluation in NLP by Gradient-Driven Optimization. ICLR
> [Wang’20] Wang et al. InfoBERT: Improving Robustness of Language Models from An Information Theoretic Perspective. ICLR
> [Zhu’20] Zhu et al. FreeLB: Enhanced Adversarial Training for Natural Language Understanding. ICLR
> [Li’21] Li et al. TAVAT: Token-Aware Virtual Adversarial Training for Language Understanding. AAAI
> [Devlin’19] Devlin et al. Bert: Pre-training of deep bidirectional transformers for language understanding. NAACL-HLT
>
>
>
> > In the network design section, it is mentioned that Theorem 1 would shed light on constraining weight norms for self-attention. It would be nice to see a concrete use case. For example, given a particular quadruple $(W^K, W^Q, W^V, W^O)$ and an input $X$, could we ablate on each weight individually and use the Theorem 1 to predict the local sensitivity?
>
> We can certainly use Theorem 1 to predict local sensitivity, and agree that ablating the weight as suggested by the reviewer could give useful insight. These new experiments are presented in Appendix C.1, Figure 3. The ablation was already partially carried out in our CIFAR10 experiments of Figure 2, where we display the results for different spectral norm constraints on $(W^V, W^O)$.  In this additional experiment, we consider an experiment very similar to your suggestion. We take a  tuple $(W^Q, W^K, W^V, W^O, X)$ from the first layer of a trained ViT on CIFAR10 and a normalized input. We then sample perturbations of each element while keeping the others fixed, observing how our local upper-bound is affected with increasing parameter perturbation size. In addition we have also included an extended study on the tightness of our bound similar to Figure 1, but we vary the input norms found in Appendix C, Figure 4. In practice, it is possible to choose the norms of $(W^K, W^Q, W^V, W^O)$ to control expansion of the of each term in our local bound throughout each layer, and we can directly examine these values in the design process (This is how we designed our networks and determined the appropriate pre-attention layer projections).

---

> > ### Comment · Reviewer_d9ub · 2023-11-23
> > **Thank you for your detailed response and changes!**
> >
> > **A1**
> > Thank you for adding the Figure 4! This figure looks great and makes the conservatism argument stronger in practice.
> >
> > **A2**
> > Thank you for adding the BERT on the SST experiment! This experiment looks great and it strengthens the applicability of the derived bounds across domains.
> >
> > **A3**
> > Thank you for adding the Figure 3! This is a really practical ablation study but it would be even better to include a value derived from Theorem 1 as an upper bound shown in the figure. It would be informative to see these curves are quite close to their Theorem 1 upper bound.
> >
> > Overall, I believe that this is a good paper and I have bumped my score to accept.

---

> > > ### Author Response · Authors · 2023-11-23
> > > **Clarification of Figure 3.**
> > >
> > > First of all, thank you for taking all of our responses into consideration.
> > >
> > > Regarding **A3**, we want to clarify that the values in Figure 3 do correspond the upper-bound given by Theorem 1 for the entire multi-head attention layer (a bound can be computed for each sampled set of parameters). However, we do think it would be useful to also plot the PGD lower bound for each sample which would give more information about tightness of the bound under these parameter perturbations. We will revise the figure to more clearly communicate its purpose.

---

### Author Response · Authors · 2023-11-22
**General Response**

Dear Reviewers,

Thank you once again for your comments and feedback. We would like to kindly remind you that we are approaching the end of the rebuttal period, so please don’t hesitate to reach out if you still have any questions or comments about our work. We are committed to addressing all concerns related to our work. Below we summarize the main points we have addressed based on your requests:
* We have strengthened our numerical experiments: We improved the study on the tightness of our derived bound by comparing to additional naive estimates (Section 5 Fig 1) and adding several new parameter ablation studies to Appendix C.1-2 (Fig 3 and 4). We have also expanded the application of our bound to a language sentiment analysis task (Stanford Sentiment Tree-bank task), showing certified robustness results on the word-embedding space as a proof of concept. These results are added to Appendix C.3 Fig 5.
* We have shown that our derived local bound is different and much tighter than a Lipschitz-based bound obtained through upper-bounding the gradient in a local region about the input (See the table provided in our response to reviewer VymQ). Thus, our derivation is practically useful to quantitatively predict robustness of dot-product self attention and *compute non-vacuous $\ell_2$ certified robustness*.
* We have added a clarifying remark to Section 4 on how our bound should be interpreted and used in design of self-attention layers to trade-off robustness and performance.

---

### Meta-Review · Area_Chair_Cmy9 · 2023-12-12

**Metareview:**

The paper studies fine-grained local sensitivity analysis of the standard dot-product self-attention. It is know that dot-product self-attention is not globally Lipschitz. This paper develops theoretical local bounds quantifying the effect of input feature perturbations and shows the role of attention weight matrices and the unperturbed input.

**Justification For Why Not Higher Score:**

More discussion on the technical novelty and the computational efficiency of the approach will be valuable to appreciate the full contributions of the paper.

**Justification For Why Not Lower Score:**

Local sensitivity of standard dot-product self-Attention is not well studied and I believe the paper has interesting technical contributions which could be valuable from both theoretical & practical perspectives.

---

### Decision · Program_Chairs · 2024-01-16

Reject